

# COVID-19 mortality is associated with pre-existing impaired innate immunity in health conditions

Matthew Lee[1], Yung Chang[2], Navid Ahmadinejad[1], Crista Johnson-Agbakwu[3], Celeste Bailey[3] and Li Liu[1,4]

[1] College of Health Solutions, Arizona State University, Phoenix, AZ, United States
[2] School of Life Sciences, Arizona State University, Tempe, AZ, United States
[3] Valleywise Health Medical Center, Phoenix, AZ, United States
[4] Biodesign Institute, Arizona State University, Tempe, AZ, United States

Corresponding author
Li Liu, liliu@asu.edu

## ABSTRACT

COVID-19 can be life-threatening to individuals with chronic diseases. To prevent severe outcomes, it is critical that we comprehend pre-existing molecular abnormalities found in common health conditions that predispose patients to poor prognoses. In this study, we focused on 14 pre-existing health conditions for which increased hazard ratios of COVID-19 mortality have been documented. We hypothesized that dysregulated gene expression in these pre-existing health conditions were risk factors of COVID-19 related death, and the magnitude of dysregulation (measured by fold change) were correlated with the severity of COVID-19 outcome (measured by hazard ratio). To test this hypothesis, we analyzed transcriptomics data sets archived before the pandemic in which no sample had COVID-19. For a given pre-existing health condition, we identified differentially expressed genes by comparing individuals affected by this health condition with those unaffected. Among genes differentially expressed in multiple health conditions, the fold changes of 70 upregulated genes and 181 downregulated genes were correlated with hazard ratios of COVID-19 mortality. These pre-existing dysregulations were molecular risk factors of severe COVID-19 outcomes. These genes were enriched with endoplasmic reticulum and mitochondria function, proinflammatory reaction, interferon production, and programmed cell death that participate in viral replication and innate immune responses to viral infections. Our results suggest that impaired innate immunity in pre-existing health conditions is associated with increased hazard of COVID-19 mortality. The discovered molecular risk factors are potential prognostic biomarkers and targets for therapeutic intervention.

## INTRODUCTION

COVID-19 was declared a global pandemic by the World Health Organization (WHO) as of March 11, 2020 (*World Health Organization, 2021a*). The outbreak has seen over 264 million cases and over 5.2 million deaths as of December 2021, and these numbers are still rising (*World Health Organization, 2021b*). The clinical spectrum of illness ranges

from asymptomatic or mild infection to severe pneumonia and death. Well documented risk factors for COVID-19 severity and fatality include age, sex, race, social determinants, and pre-existing health conditions (*Takagi, 2021*; *Centers for Disease Control and Prevention, 2020*; *Jin et al., 2020*; *Makary, 2020*; *Williamson et al., 2020*). The fatality rate stratified by age groups steadily increases from 0.2 per 100,000 patients in children (<14 years old) to 1,797.8 in elders (≥85 years) (*Centers for Disease Control and Prevention, 2020*). Age-adjusted fatality rate in men is 1.4 times higher than in females (*Jin et al., 2020*). More importantly, people with pre-existing health conditions are susceptible to extreme outcomes. While diagnosis rates of COVID-19 are nearly equal for patients with and without comorbidities, those with a comorbidity account for 83.29% of COVID-19 deaths (*Makary, 2020*). Not all comorbidities have the same impact on COVID-19 prognosis. By linking primary care records of >17 million adults to 10,926 COVID-19 related deaths, the OpenSAFELY project estimated age-and-sex-adjusted hazard ratios (HR) of COVID-19 related deaths for 23 groups of pre-existing health conditions (*Williamson et al., 2020*). It reported that patients with organ transplant were at the highest risk (HR = 6.00) and those with high blood pressure were at the lowest risk (HR = 1.09). However, the biological mechanisms underlying such distinct impacts are largely unknown, which impedes the development of effective interventions to improve clinical outcomes.

Independently, mechanistic studies of COVID-19 pathogenesis have found that dysregulated biological processes in hosts play an important role in disease severity. It is widely recognized that COVID-19 patients with cytokine storms have high mortality (*Hojyo et al., 2020*). Serum levels of inflammatory factors, such as interleukins and C-reactive protein, have been proposed as prognostic markers (*Benard et al., 2021*). Early metabolic responses to infections also show signature differences between patients with favorable outcomes and patients with unfavorable outcomes (*Xiao et al., 2021*; *Ayres, 2020*). Given that comorbidities and molecular dysregulations influence COVID-19 severity, we must question what molecular abnormalities associated with pre-existing conditions predispose COVID-19 patients to poor prognosis and to what extent.

To answer this question, an intuitive approach would be to examine molecular profiles of COVID-19 patients before SARS-CoV-2 infection and correlate with their prognoses after infection. However, this strategy is not feasible because before-infection samples of COVID-19 patients are rarely collected. To circumvent this obstacle, we propose a novel approach using data from disjoint cohorts, in which molecular dysregulations in pre-existing conditions are linked to COVID-19 prognosis at the health condition level, making use of summary statistics from epidemiology studies.

Given that several comorbidities increase the COVID-19 mortality rate, we hypothesize that these comorbidities share common molecular dysregulations prior to SARS-CoV-2 infection, and the magnitude of such dysregulations contributes to the severity of COVID-19 outcome. To test this hypothesis, we need quantitative data of COVID-19 mortality rates and molecular profiles obtained from people with various health conditions, which fortunately are readily available. Specifically, the OpenSAFELY study has published the HRs of COVID-19 mortality for 23 groups of pre-existing conditions

(*Williamson et al., 2020*). Molecular profiles of individuals affected by these conditions and those unaffected can be found in public repositories, such as the Gene Expression Omnibus (GEO) database (*Barrett et al., 2013*). Because these data were generated before the pandemic, all samples were COVID negative. It is widely acknowledged that patient gene expression profiles reflect the underlying pathological processes (*Kurreck & Stein, 2015*). While different diseases target different tissues and organs, transcriptomes of peripheral blood cells are informative about systematic changes of a person's overall health (*Mesko et al., 2010*; *Aziz, Zaas & Ginsburg, 2007*). Therefore, we chose to examine peripheral blood transcriptomes in this study. We identify transcriptional dysregulations observed in multiple pre-existing conditions and correlate their fold changes with HRs of COVID-19 mortality. These recurrent dysregulations correlated with HRs are molecular risk factors predisposing COVID-19 patients to severe outcomes. We further analyzed functional relationships of these risk genes, which converged onto impaired innate immunity as a systemic mechanism that weakens host defense against SARS-CoV-2 infection.

While we focused on gene expression risk factors in this study, our analytic approach is applicable to other omics-level profiles, such as epigenetic and metabolomic data. Integration of these discoveries will allow for better prediction of severe outcomes of COVID-19 and inform the development of preventative measures to reduce fatality. Furthermore, the long-term sequelae of COVID-19 survivors are currently unknown. A greater apprehension of the disease mechanisms in the context of comorbidities will serve for future evaluation of the health impact of COVID-19 on patients with chronic diseases.

## MATERIALS AND METHODS

### Data sets

The OpenSAFELY project reported age-sex-adjusted HRs of COVID-19 related deaths for 23 groups of pre-existing health conditions. We excluded the two cancer groups (solid tumors and hematological malignancies) due to the extremely high heterogeneity of cancers (*Chandrashekar et al., 2020*). For each remaining health condition, we searched the GEO database (*Barrett et al., 2013*) to identify transcriptomics studies involving affected individuals (cases) and unaffected individuals (controls, Fig. 1A). We limited our queries to peripheral blood samples as a *modus operandi* of removing confounders related to different tissue types and encapsulating disease characteristics at a systemic level. We further limited our query to microarray-based transcriptomic profiles to reduce technical variance. If multiple data sets were available for a health condition, we chose the one with the largest sample size. We downloaded the normalized gene expression values.

### Identify dysregulated gene expression in pre-existing health conditions

Given a transcriptomic data set, we used the Student's *t*-test to compare expression levels of each gene in cases *vs* controls (Fig. 1B). Specifically, we used the t.test() function in R and set the following parameters (alternative="two.sided", paired=F). Aiming to be inclusive at this step, we considered genes with nominal $P$ value $<0.05$ to be differentially

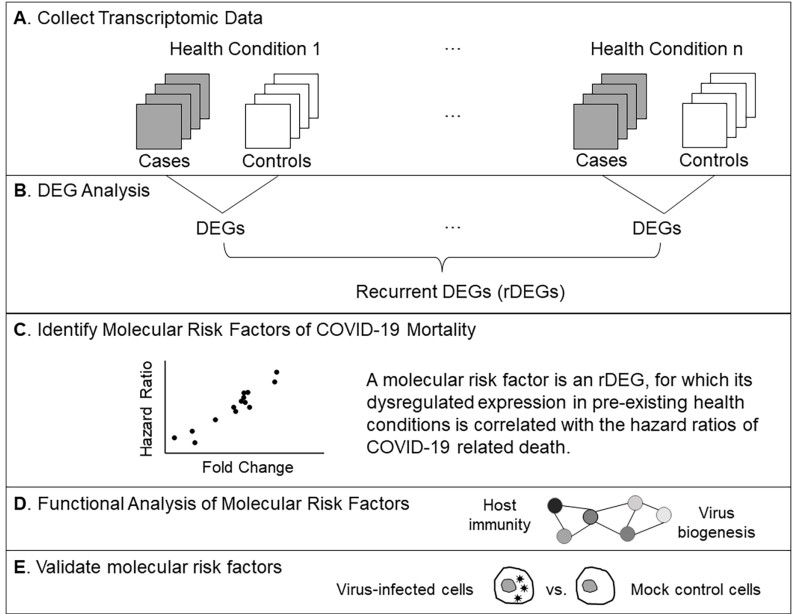

**Figure 1 Analysis workflow.** (A) Compile microarray-based transcriptomic data sets for common health conditions. We included case-control studies using peripheral blood samples. (B) Find differentially expressed genes (DEGs) for each health condition. Recurrent DEGs were those differentially expressed in at least four health conditions. (C) Perform correlation tests to identify molecular risk factors, which are pre-existing expression dysregulations that increase the risk of COVID-19 related death. (D) Examine functional relationships of molecular risk factors *via* enrichment and network analyses. (E) Validate molecular risk genes using published data from an *in vitro* SARS-CoV-2 infection study (*Chiu, Macmillan & Chen, 2009*). 

expressed. If multiple probes on the microarray represented the same gene, we kept the one with the lowest *P* value and removed the other ones to avoid redundancy. For a differentially expressed gene (DEG), we computed the fold change (FC) as the ratio of the mean expression level in cases over controls. For a non-DEG, we set the FC to one. If a gene was differentially expressed in at least four health conditions, it was designated a recurrent DEG (rDEG).

## Identify molecular risk factors of COVID-19 mortality

The gene expression data and the COVID-19 data were derived from different cohorts. To perform cross-cohort analysis, we linked FCs of gene expression and HRs of COVID-19 mortality at the health condition level. For a given rDEG, we tested if its FCs in pre-existing conditions were correlated with the corresponding HRs using the Pearson correlation test (Fig. 1C). Specifically, we used the cor.test() function in R and set the following parameters (method="pearson", use="pairwise.complete.obs"). We corrected for multiple comparisons by converting nominal *P* values to false discovery rates (FDRs) using the Benjamini-Hochberg method, as implemented in the p.adjust() function in R. FDR < 0.05 indicated significant molecular risk factors. FDR > 0.05 but a nominal *P* value <0.01 indicated marginal risk factors. The positive or negative sign of a Pearson correlation coefficient (PCC) indicated that upregulated gene expression or downregulated gene

expression in pre-existing conditions increased the risk of COVID-19 mortality, respectively.

## Functional categorization and analysis

We classified molecular risk factors into overlapping gene sets based on annotations of biological processes in the Gene Ontology (GO) database and pathways in the KEGG database. For each gene set, we tested if it was overrepresented using the Fisher's exact test. We corrected for multiple comparisons by converting nominal $P$ values to FDRs. We built association networks of enriched gene sets (FDR < 0.05) and examined their relationships using the cnetplot() function from the R/enrichplot package (*Yu et al., 2012*) (Fig. 1D). To assess the power of the association network, we performed hierarchical clustering of the gene sets. Specifically, we constructed an input matrix with genes as rows and GO terms as columns. If a gene is annotated with a GO term, the corresponding value in this matrix is set to 1. Otherwise, it is set to 0. We then performed hierarchical clustering and computed the Goodman-Kruskal-gamma index (*Baker & Hubert, 1975*) for various numbers of clusters using the NbCluster method (distance="binary", index="gamma", method="ward.D2", max.nc=9) (*Charrad et al., 2014*).

## Experimental validations

To validate if the identified molecular risk factors at the gene level reflected the molecular changes at the protein level observed in SARS-CoV-2 infections, we used the published proteomic data set from an *in vitro* infection experiment (*Bojkova et al., 2020*). In this experiment, human Caco-2 cells were infected with SARS-CoV-2 viruses. Infected cells and mock controls, each with three replicates, were cultured for 24 h. Quantifications of 6,381 proteins were obtained and compared between the infected cells and the mock control cells with two-group $t$ tests. We obtained the normalized protein quantification data and the pre-computed $t$-test $P$ values from the Table S2 in the published study (*Bojkova et al., 2020*). Given a risk factor gene, we searched the proteomics data set for the matching protein based on the gene symbol. To validate a risk factor gene, we required that its transcriptional dysregulation pattern was consistent with the dysregulated protein expression pattern (Fig. 1E). Specifically, if upregulation of the risk factor gene was associated with an increased COVID-19 mortality rate, the matching protein shall also be upregulated in SARS-CoV-2 infected cells as compared to mock control cells. If downregulation of the risk factor gene was associated with an increased COVID-19 mortality rate, we required the matching protein also to be downregulated in SARS-CoV-2 infected cells as compared to mock control cells.

R source codes used in this study are available at https://github.com/liliulab/COVID19_Mortality_Association/.

## RESULTS

### DEGs in pre-existing health conditions

Our search of the GEO database found qualified transcriptomic data for 14 health conditions. For each health condition, we identified DEGs with Student $t$-test $P < 0.05$.

**Table 1 Data sets and DEGs of 14 pre-existing condition.**

| Health condition | Gene expression | | | COVID-19 mortality* | |
|---|---|---|---|---|---|
| | GEO accession | DEG count | Mean FC | Disease group | HR |
| Alzheimer's disease | GSE63060 | 5,413 | 0.99 | Stroke or dementia | 2.57 |
| Asthma | GSE110551 | 1,215 | 1.01 | Asthma | 1.55 |
| Coronary artery disease | GSE12288 | 1,289 | 1.10 | Chronic heart disease | 1.57 |
| Chronic obstructive pulmonary disease | GSE42057 | 2,054 | 0.99 | Respiratory disease excluding asthma | 1.95 |
| Hispanic ethnicity | GSE30101 | 4,249 | 1.13 | Ethnicity | 1.37 |
| HIV | GSE104640 | 11,503 | 1.00 | Immunosuppressive condition | 2.75 |
| Hypertension | GSE135111 | 413 | 1.53 | High blood pressure or diagnosed hypertension | 1.09 |
| Lupus | GSE37356 | 2,289 | 1.00 | Rheumatoid arthritis, lupus or psoriasis | 1.30 |
| Obesity | GSE110551 | 3,905 | 1.00 | BMI > 35 | 1.81 |
| Rheumatoid Arthritis | GSE93272 | 9,902 | 1.01 | Rheumatoid arthritis, lupus or psoriasis | 1.30 |
| Type-2 Diabetes | GSE65561 | 4,155 | 0.99 | Diabetes | 2.27 |
| Chronic Kidney Disease | GSE37171 | 17,688 | 1.01 | Reduced kidney function (eGFR < 30) | 3.48 |
| Multiple sclerosis | GSE21942 | 7,735 | 1.12 | Other neurological disease | 3.08 |
| Alcoholic Hepatitis | GSE28619 | 9,489 | 1.05 | Liver disease | 2.39 |

**Note:**
* as reported in the OpenSAFELY Project (*Williamson et al., 2020*).

This lenient cutoff allowed us to be as inclusive as possible at this step. On average, each health condition was associated with 5,777 DEGs (range 1,215 to 17,688). Most of the DEGs were downregulated in cases as compared to controls (mean FCs range from 0.003 to 0.160). Among a total of 25,552 genes analyzed, we found 11,930 rDEGs, that is, those that were differentially expressed in at least four health conditions. Table 1 presents the summary statistics of DEGs.

## Pre-existing expression dysregulations increase COVID-19 death risks

For each rDEG, we tested if its FCs in different health conditions were correlated with HRs for COVID-19 mortality. For health conditions where this gene was not differentially expressed, we set the FCs to 1 and included them in the correlation test as well. Among a total of 11,930 DEGs, we found no significant molecular risk factor that passed the stringent FDR < 0.05 threshold. However, 231 genes passed the Pearson correlation test $P < 0.01$ threshold and were considered as marginal molecular risk factors. Among them, upregulated expression of 70 genes and downregulated expression of 181 genes increased risk of COVID-19 related death (Table S1).

The *RPS28* gene had the most significant correlation $P$ value (0.0003). Its FCs in pre-existing conditions were positively correlated with HRs of COVID-19 mortality (PCC = 0.83, Fig. 2A). *RPS28* encodes a component of the 40S subunit of the ribosome where a cell synthesizes proteins. It was differentially expressed in six health conditions, including rheumatoid arthritis, chronic obstructive pulmonary disease, alcoholic hepatitis, multiple sclerosis, HIV, and chronic kidney disease. As its FC increased from 0.98 to 1.14, the HR of COVID-19 mortality increased from 1.30 to 3.48. Furthermore, the

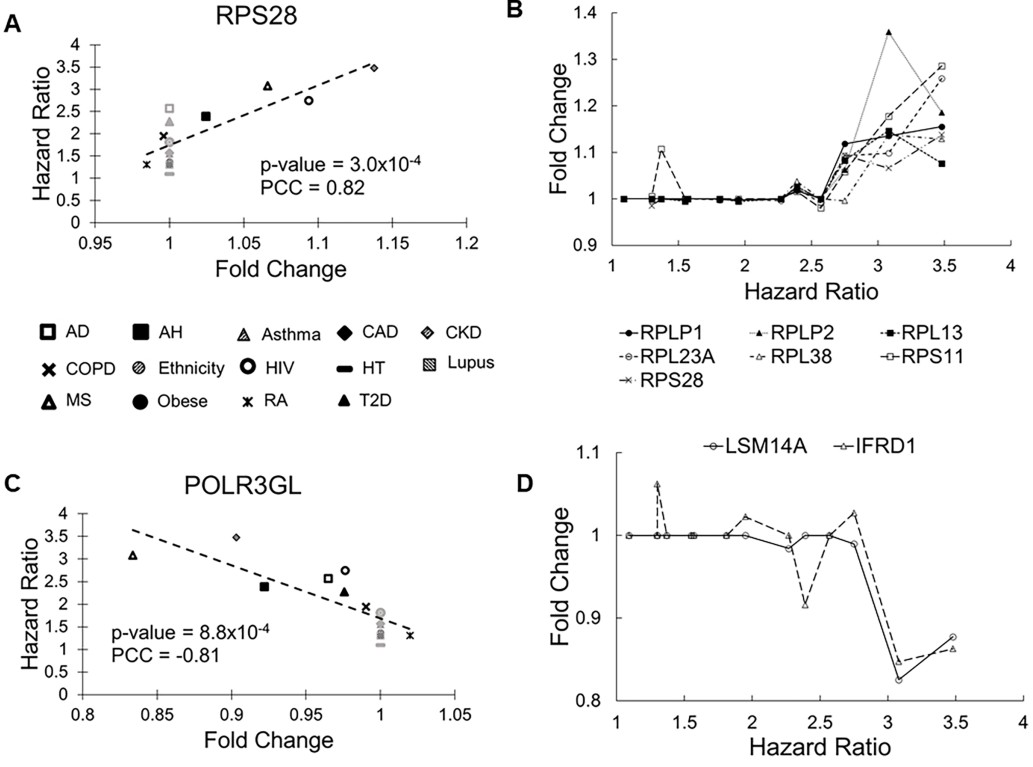

**Figure 2 Scatter plots of selected genes showing correlations between fold change (FC) and hazard ratio (HR).** (A) *RPS28* had the lowest Pearson correlation test *P* values and positive PCC values. The 14 health conditions are represented by different symbols (AD, Alzheimer's disease; AH, alcoholic hepatitis; Asthma, asthma; CAD, coronary artery disease; CKD, chronic kidney disease; COPD, chronic obstructive pulmonary disease; Ethnicity, Latino *vs* Caucasian; HIV, human immunodeficiency virus; HT, hypertension; Lupus, lupus; MS, multiple sclerosis; Obese, obesity; RA, rheumatoid arthritis; and T2D, type-2 diabetes). Black symbols indicate health conditions in which a given gene was differentially expressed. Gray symbols indicate health conditions in which a given gene was not differentially expressed. Broken lines represent linear fits between FC and HR. (B) Seven genes coding ribosomal proteins consistently show positive correlations between expression FCs and HRs of COVID-19 mortality. (C) *POLR3GL* had the lowest Pearson correlation test *P* values and negative PCC values. It is involved interferon production, providing anti-viral innate immunity. (D) Two additional genes involved in interferon signaling show negative correlations between FC and HR.

list of molecular risk factors contained seven additional genes that encode ribosomal components (*RPLP1*, *RPLP2*, *RPL13*, *RPL23A*, *RPL30*, *RPL38*, and *RPS11*). Except for *RPL30*, upregulation of these genes consistently increased the HR of COVID-19 mortality (*P* range = 0.001 to 0.009, PCC range = 0.71 to 0.78, Fig. 2B). This is in accordance with the positive viral infection-regulating roles of ribosomal proteins (*Li, 2019*). Notably, three ribosomal proteins in our list are required for early virus accumulation (*Campos et al., 2016*), though mechanistic studies in SARS-CoV-2 are still lacking.

The *POLR3GL* gene showed the most significant negative correlation (*P* = 0.0008, PCC = −0.81, Fig. 2C). *POLR3GL* encodes a subunit of RNA polymerase III that catalyzes the transcription of DNA into RNA. It induces production of interferon (IFN-α/β) to inhibit virus replication (*Chiu, Macmillan & Chen, 2009*; *Samuel, 2001*). Consistent with this function, pre-existing downregulation of *POLR3GL* in eight health conditions

**Table 2 Significantly enriched gene sets.**

| ID | Description | Enrich | FDR | Count | Genes |
|----|-------------|--------|-----|-------|-------|
| *KEGG pathways* | | | | | |
| hsa03010 | Ribosome | 4.77 | 0.00026 | 8 | *RPL38/RPS11/RPS28/RPLP1/RPLP2/RPL13/RPL30/RPL23A* |
| hsa00563 | GPI-anchor biosynthesis | 10.87 | 0.0025 | 3 | *PIGP/PIGA/PIGG* |
| hsa05171 | Coronavirus disease COVID-19 | 3.25 | 0.0031 | 8 | *RPL38/RPS11/RPS28/RPLP1/RPLP2/RPL13/RPL30/RPL23A* |
| *Gene Ontology biological process* | | | | | |
| GO:0045047 | protein targeting to ER | 8.55 | 0.00036 | 10 | *RPL38/RPS11/SEC62/RPS28/RPLP1/RPLP2/SRP9/RPL13/SGTB/RPL23A* |
| GO:0006613 | co-translational protein targeting to membrane | 8.33 | 0.00087 | 9 | *RPL38/RPS11/SEC62/RPS28/RPLP1/RPLP2/SRP9/RPL13/RPL23A* |
| GO:0022626 | cytosolic ribosome | 7.34 | 0.0040 | 8 | *RPL38/RPS11/RPS28/RPLP1/RPLP2/RPL13/RPL30/RPL23A* |
| GO:0006413 | translational initiation | 5.23 | 0.0055 | 10 | *HABP4/RPL38/RPS11/RPS28/RPLP1/RPLP2/RPL13/PAIP2/COPS5/RPL23A* |
| GO:0019083 | viral transcription | 5.13 | 0.013 | 9 | *RPL38/RPS11/RPS28/RPLP1/RPLP2/RPL13/NUP160/POLR2C/RPL23A* |
| GO:00116607 | nuclear speck | 3.45 | 0.014 | 13 | *PNN/HABP4/CBLL1/HBP1/ACADM/SRSF2/SRSF7/RFXAP/DDX42/SART1/POLDIP3/NRIP1/SAP18* |
| GO:0000184 | nuclear-transcribed mRNA catabolic process | 5.89 | 0.031 | 7 | *RPL38/RPS11/RPS28/RPLP1/RPLP2/RPL13/RPL23A* |
| GO:0008380 | RNA splicing | 3.01 | 0.032 | 14 | *PNN/HABP4/DHX8/ZRANB2/HNRNPH3/HNRNPA2B1/ZNF326/SRSF2/SRSF7/DDX42/PPWD1/SART1/POLR2C/SAP18* |
| GO:0044391 | ribosomal subunit | 4.32 | 0.040 | 8 | *RPL38/RPS11/RPS28/RPLP1/RPLP2/RPL13/RPL30/RPL23A* |
| GO:0031307 | mitochondrial outer membrane component | 15.40 | 0.049 | 3 | *SYNJ2BP/TOMM20/BNIP3* |

(Alzheimer's disease, chronic kidney disease, alcoholic hepatitis, chronic obstructive pulmonary disease, HIV, multiple sclerosis, rheumatoid arthritis, and type-2 diabetes) increased the risk of COVID-19 related death. Furthermore, the list of risk factors contained two additional genes, *LSM14A* and *IFRD1*, that regulate interferon signaling. For these genes, downregulation increased HRs of COVID-19 mortality ($P = 0.003$ and 0.005, PCC = −0.74 and −0.71, respectively, Fig. 2D). Interestingly, we did not find interferons as molecular risk factors, presumably due to their transient expression profiles.

## Functional groups enriched with risk factors

We classified the list of marginal molecular risk factors into functional gene sets based on Gene Ontology and KEGG annotations and then performed enrichment analysis. At FDR < 0.05, these molecular risk factors were significantly enriched in 10 biological processes and three pathways (Table 2). Most of these gene sets were related to viral transcription, mRNA processing and metabolism, protein synthesis, and endoplasmic reticulum (ER) function.

We then built an association network to examine functional relationships of the enriched gene sets (Fig. 3A). Based on hierarchical clustering analysis and Goodman-Kruskal-gamma index, genes in this association network formed seven clusters (gamma = 1.0, Fig. S1A). Merging these clusters *via* GO terms produced three supersets

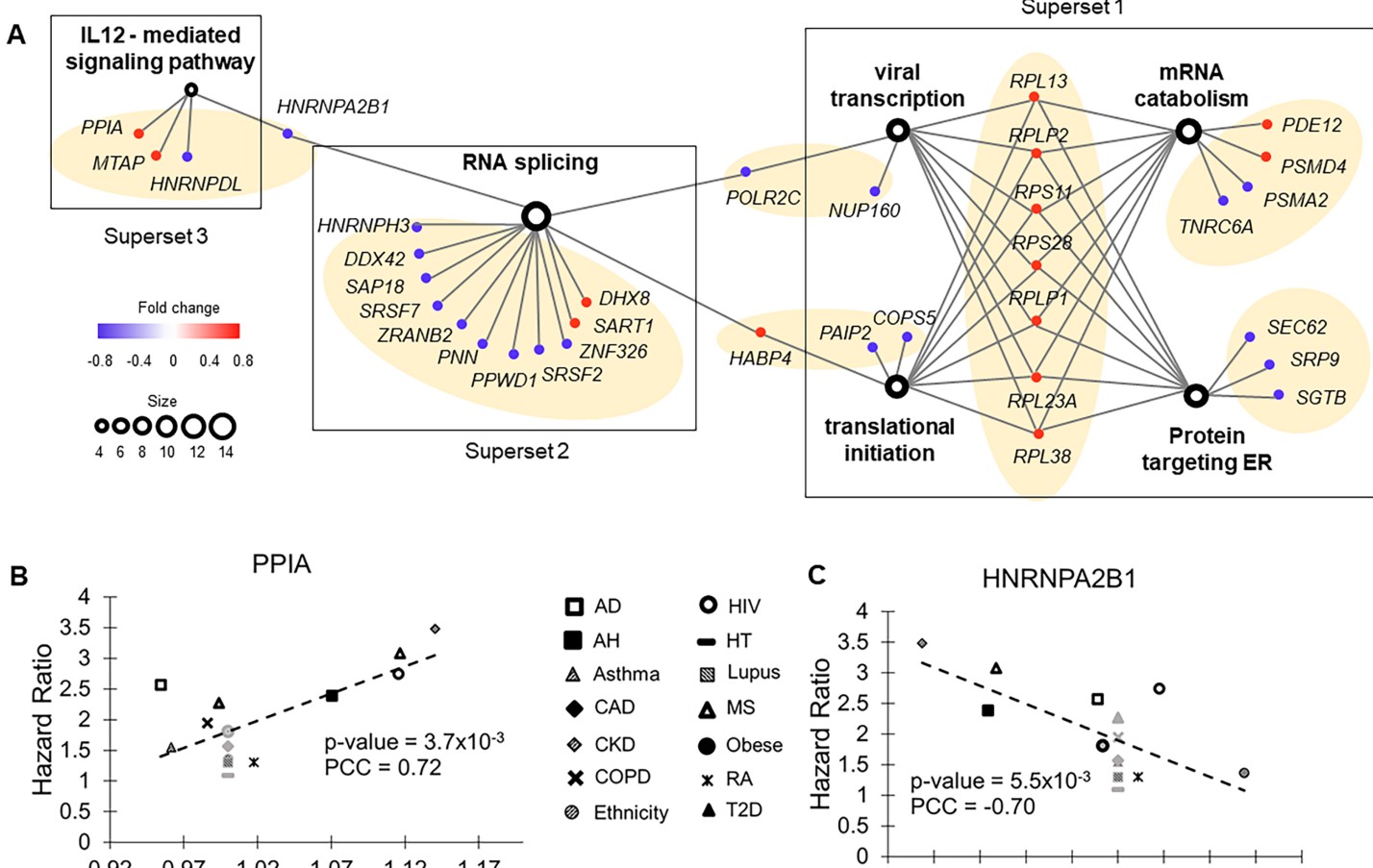

**Figure 3 Functional groups enriched with molecular risk factor genes.** (A) The association network of gene sets with FDR < 0.05 from enrichment analysis. This network contained two types of nodes, with gene sets represented by open circles and individual risk factor genes by colored dots. An edge links a risk factor gene to its associated gene set. The size of an open circle is proportional to the number of risk factor genes connected to it. The color of a dot indicates if its upregulation is positively (red) or negatively (blue) correlated with HR of COVID-19 mortality. The seven gene clusters are highlighted with light orange background. (B) Upregulation of *PPIA* increased risk of COVID-19 mortality. *PPIA* facilitates virus replication. (C) Downregulation of *HNRNPA2B1* increased risk of COVID-19 mortality. *HNRNPA2B1* inhibits virus replication.

(gamma = 0.84, Fig. S1B), each composed of interconnected gene clusters. Genes bridging these sets indicate crosstalk between supersets.

The first superset consisted of gene clusters involved in viral transcription, translational initiation, mRNA catabolic process, and proteins targeting ER. It included eight common risk factor genes encoding ribosomal components that function in ER. Except *RPL30*, upregulation of all these genes correlated with increased HRs of COVID-19 mortality. Conversely, for most of the other genes (7 out of 9) in this superset, downregulation correlated increased HRs of COVID-19 mortality, including two genes (*SEC62* and *SGTB*) that target ER and participate in degradation of misfolded proteins. Therefore, pre-existing abnormal functions in ER, specifically upregulated protein synthesis and

downregulated degradation of unfolded protein, were associated with high risk of COVID-19 death.

The second superset consisted of a single gene cluster involved in RNA splicing. For almost all genes (11 out of 14) in this set, downregulation was associated with increased HRs of COVID-19 mortality. In COVID-19 patients, host RNA splicing was significantly disrupted by SARS-CoV-2 (*Banerjee et al., 2020*). Our observation suggests that pre-existing downregulation of RNA splicing genes can potentially aggravate such disruptions.

The third superset consisted of a single gene cluster participating in the interleukin-12-mediated signaling pathway. Noticeably, one of the risk factor genes in this set, *PPIA*, has been shown to act as a potential mediator between human SARS coronavirus nucleoprotein and BSG/CD147 during the process of invasion of host cells by the virus (*Chen et al., 2005*). Consistent with this previous study, we found that pre-existing upregulation of this gene increased COVID-19 mortality risk in nine common health conditions (Fig. 3B). Interestingly, interleukin-12, a pro-inflammatory cytokine (*IL12A* and *IL12B* genes) was not a molecular risk factor. It was dysregulated in two health conditions but did not meet the criterion of being a rDEG, which required dysregulation in at least four health conditions. Its receptor *IL12RB1* was a near miss, showing a positive correlation between FC and HR in five health conditions ($P = 0.079$, PCC = 0.48).

Crosstalk between the second and third supersets is *via HNRNPA2B1*, which binds heterogeneous nuclear RNA (hnRNA) and subsequently induces IFN-$\alpha$/$\beta$ production to inhibit virus replication.

Two other hnRNA binding proteins, *HNRNPH3* and *HNRNPDL*, were also molecular risk factors. For all three genes, pre-existing downregulation increased COVID-19 mortality risk, presumably by blocking IFN-$\alpha$/$\beta$ production, which compromises innate immunity (Fig. 3C).

## Dysregulated cell death and mitochondrial functions

The innate immune system is intrinsically connected with programmed cell death (*Birge & Ucker, 2008*) and mitochondrial functions (*West, Shadel & Ghosh, 2011*). In accordance with this, our list of molecular risk factors contained eight genes involved in apoptosis, seven genes involved in autophagy, and 20 genes involved in mitochondrial function (Table S2). Although these gene sets did not pass the stringent threshold of FDR < 0.05 in the enrichment analysis, they were overrepresented in several biological processes with borderline nominal *P* values, including "mitochondrial RNA metabolism" ($P = 0.001$), "regulation of apoptotic signaling pathway" ($P = 0.05$), "mitochondrial organization" ($P = 0.04$), "autophagy of mitochondrion" ($P = 0.08$), and "autophagosome assembly" ($P = 0.08$).

To examine how these dysregulated processes correlated to COVID-19 mortality, we built another association network (Fig. 4A). Autophagy and apoptosis are two mechanisms of programmed cell death that inhibit each other (*Thorburn, 2008*). Our association network contains six genes that co-regulate these processes. In normal conditions, these genes keep autophagy and apoptosis in balance. Specifically, *BNIP3*, *FBXW7*, *RAB7A*,

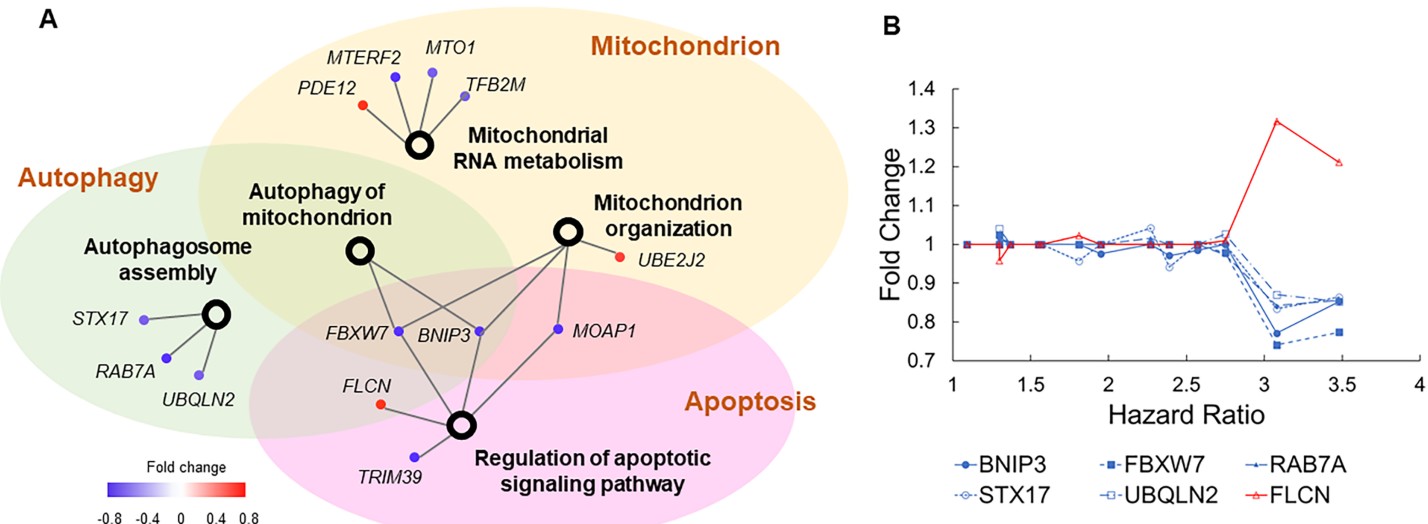

**Figure 4 Genes involved in autophagy, apoptosis, and mitochondrial functions.** (A) The association network of gene sets with nominal *P* values <0.1 from enrichment analysis. This network contained two types of nodes, with gene sets represented by open circles and individual risk factor genes by colored dots. An edge links a risk factor gene to its associated gene set. The color of a dot indicates if its upregulation is positively (red) or negatively (blue) correlated with HR of COVID-19 mortality. (B) Five genes (blue lines) inducing autophagy and suppressing apoptosis show negative correlations between expression FCs and HRs of COVID-19 mortality. One gene (red line) suppressing autophagy and inducing apoptosis shows positive correlations between expression FCs and HRs of COVID-19 mortality.

*STX17*, and *UBQLN2* induce autophagy and suppress apoptosis, which is counteracted by *FLCN* (*Khandia et al., 2019*; *Possik et al., 2014*). Pre-existing dysregulations of these genes converged to a common pattern, *i.e.*, suppressed autophagy and escalated apoptosis jointly increased the risk of COVID-19 related death (Fig. 4B). Furthermore, downregulation of *BNIP3* and *FBXW7* suggest compromised mitochondrial organization and reduced mitochondrial autophagy. As disrupted mitochondrial functions, suppressed autophagy, and escalated apoptosis are commonly found in COVID-19 patients (*Ajaz et al., 2021*; *Gassen et al., 2021*; *Paolini et al., 2021*), our results imply if such dysregulations are present prior to SARS-CoV-2 infection, patients are prone to poor prognosis.

## Validation in SARS-CoV2 infected samples

We validated the identified risk factor genes using a published proteomics data set from an *in vitro* study (*Bojkova et al., 2020*). In this experiment, human Caco-2 cells were infected with SARS-CoV-2 viruses. Infected cells and mock controls, each with three replicates, were cultured for 24 h. Quantifications of 6,381 proteins were obtained. We cross-referenced these proteins with the molecular risk factors based on gene symbols and found 105 gene-protein pairs. For each pair, we first examined if the protein had significantly different expression levels in infected cells compared to mock control cells using the published pre-computed *t*-test *P* value <0.05 as the threshold. We then examined if the protein dysregulation pattern was consistent with the dysregulation patterns of the risk factor genes, which would exacerbate the pre-existing aberrations and lead to poor prognosis.

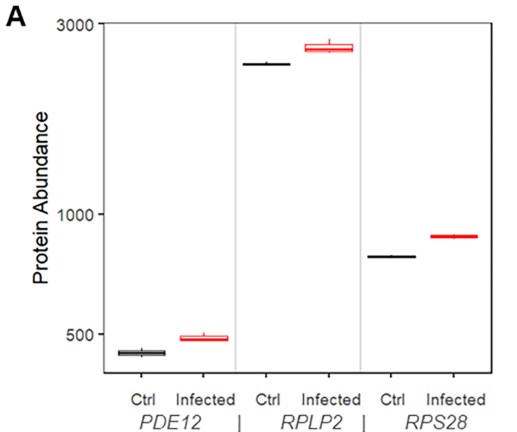
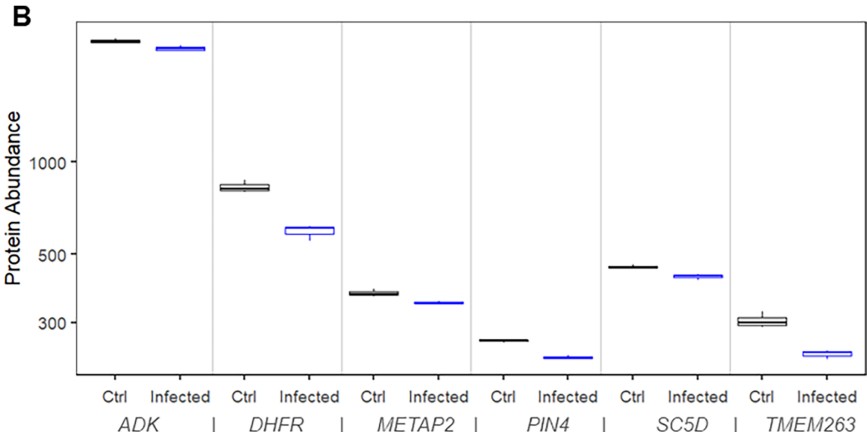

**Figure 5 Protein expression of risk factor genes. Boxplot of protein abundance (log scale) in mock controls and SARS-CoV-2 infected cells.** (A) The plot includes genes for which upregulation increased risk of COVID-19 related death. (B) The plot includes genes for which downregulation increased risk of COVID-19 related death.

We found that 11 proteins were significantly upregulated in the infected cells compared to mock control cells. Among them, three (*RPS28*, *RPLP2*, and *PDE12*) had matching risk factor genes showing consistent patterns, *i.e.*, transcriptional upregulations increased the COVID-19 mortality risk (Fig. 5A). At 24 h after infection, the expression level of these proteins was significantly higher than that in mock controls by 9.5% to 12.2% (*P* values range 0.00027 to 0.019).

Similarly, we found that 11 proteins were significantly downregulated in the infected cells compared to mock control cells. Among them, six (*ADK*, *DHFR*, *METAP2*, *PIN4*, *SC5D*, and *TMEM263*) had matching risk factor genes showing consistent patterns, *i.e.*, transcriptional downregulations increased the COVID-19 mortality risk (Fig. 5B). At 24 h after infection, the expression level of the corresponding protein was significantly lower than that in mock controls by 5.3% to 28.6% (*P* values range 0.0004 to 0.04).

In summary, we validated nine molecular risk factor genes by showing the concordance between transcriptional dysregulations caused by pre-existing medical conditions and the protein dysregulations caused by SARS-CoV-2 infection. The joint effects plausibly modulated the clinical outcomes of COVID-19 patients.

## DISCUSSION

The drastically different disease progression and prognosis among COVID-19 patients with pre-existing health conditions challenge clinical management of this life-threatening disease. In this study, we integrated publicly available transcriptomics data of common health conditions and COVID-19 epidemiology data to study the molecular mechanisms underlying this complex problem. Our analyses revealed that pre-existing transcriptional dysregulations frequently observed in multiple health conditions increased risk of severe COVID-19 outcomes, plausibly *via* impairing host innate immunity, as discussed below.

Innate immunity is an integral part of the body's defense system which responds to invading pathogens, as well as damage caused by chronic health conditions (*Horiguchi*

*et al., 2018*). Abnormalities of many biological processes may alter innate immune responses, which in turn reduces host defense against SARS-CoV-2 infection (*Schultze & Aschenbrenner, 2021*).

ER and mitochondria are crucial organelles that lie at the crossroad between host physiological functions and viral infection. On the one hand, various chronic disorders interfere with ER homeostasis and mitochondrial dynamics, giving rise to chronic inflammation that subsequently activates the body's innate immune system (*Ozcan & Tabas, 2012*; *Bhatti, Bhatti & Reddy, 2017*). On the other hand, invading viruses hijack ER for entry into host cells and assembly of viral genomes (*Inoue & Tsai, 2013*). Mitochondrial dynamics are subverted to benefit the intracellular survival of viruses. Emerging evidence suggests that coronavirus infection, including SARS and COVID-19, triggers ER stress and viral replication within mitochondrial structures (*Chan et al., 2006*; *Banerjee et al., 2020*; *Zhang et al., 2020*). If these organelles are already compromised prior to infection, further disruptions of their crucial functions by COVID-19 will likely lead to severe outcomes. Our study supports this hypothesis as we found genes targeting ER and mitochondria were enriched in the list of molecular risk factors. These genes include *SEC62* and *SGTB* that degrade unfolded proteins in ER; *UBE2J2*, *COPS5*, *MBTPS2*, and *PPIA* that respond to unfolded proteins; and 20 genes that participate in various aspects of mitochondrial functions.

Innate immune responses to virus-infected cells include autophagy to isolate and clear infected viruses or apoptosis and necroptosis to eliminate infected cells. Viruses have evolved mechanisms to inhibit these surveillance processes so that infected cells would not be cleared efficiently, and the viruses could spread (*Kvansakul, 2017*). Suppressed autophagy and escalated apoptosis have been reported in severe COVID-19 cases (*Gassen et al., 2021*; *Paolini et al., 2021*). In this study, we discovered transcriptional dysregulations that predisposed patients to such distortion. Interestingly, two of these genes, *FBXW7* and *BNIP3*, determine the cell fate *via* autophagy of mitochondria and apoptosis, implying the important role of mitochondrial functions in COVID-19 severity.

Both chronic inflammation in pre-existing health conditions and acute inflammation in pathogen infections regulate interferon production. Reduced antiviral interferon response has been associated with excessive proinflammatory responses in COVID-19 and emerges as a clinical determinant of COVID-19 severity (*Blanco-Melo et al., 2020*). Our results also imply that interferon production and signaling are suppressed in several health conditions *via* downregulation of *POLR3GL*, *LSM14A*, *IFRD1*, and three hnRNA binding proteins (*HNRNPA2B1*, *HNRNPH3*, and *HNRNPDL*). Dysregulation of these genes disrupts type I interferon signaling pathways. Furthermore, we found upregulation of *PPIA* and *MTAP* genes that activate IL12-mediated signal pathway in which proinflammatory cytokines IL12 mediate the innate immune response (*Trinchieri, 2003*). However, we did not find any cytokines, either proinflammatory or anti-inflammatory, to be molecular risk factors. Although several cytokines were dysregulated in the health conditions we examined, there was no uniform correlation with HR of COVID-19 mortality.

Limitations of our study include the lack of individual risk factors passing a stringent statistical threshold and no consideration of multivariate effects. Although our analysis identified marginal molecular risk factors passing the nominal $P$ value cutoff, none had a significant FDR after correction for multiple comparisons, which disqualified them as prognostic markers. However, analysis using the aggregation of these risk factor genes discovered significantly enriched biological processes, with the best FDR < 10-4 (Table 2). Therefore, we are confident that chronic ER stress and immune dysregulation in pre-existing health conditions increased risk of COVID-19 mortality. Our analyses were based on univariate models, in which we examined the expression levels of each gene separately. Because multiple genes are dysregulated concurrently and a combination of them contributes to COVID-19 prognosis, a more realistic model should consider their combined effect. However, because the transcriptomics data were derived from individual patients and HRs of COVID-19 mortality were from summary statistics of an epidemiology study, we chose to use univariate models that are more straightforward to interpret. Lastly, our analyses focused on associating pre-existing transcriptional dysregulations to the ultimate outcome of COVID-19 assessed on mortality. However, dynamic changes of molecular and cellular processes are expected during the clinical course of COVID-19, which are potentially influenced by pre-existing conditions, as well.

Meta-analysis studies have emerged to associate pre-existing conditions with severe COVID-19 outcomes (Treskova-Schwarzbach et al., 2021; Rosario Ferreira et al., 2021). Our study adds to the current literature by developing a new analytical approach to integrating epidemiology data and omics data derived from disjoint cohorts and discovering novel molecular risk factors. While we focus on transcriptional regulation in this study, an immediate next step is to apply this approach to other molecular changes, including genetic variation, epigenetic modification, and metabolic perturbation to investigate their roles in COVID-19 pathogenesis. As before-infection samples of COVID-19 patients are difficult to acquire, integration of existing multi-omics data and epidemiology data hold promise to accelerate the discovery of diagnostic and therapeutic markers to improve the management of COVID-19 disease.

## CONCLUSIONS

Our study illuminates that gene expression dysregulations in pre-existing health conditions that impair innate immunity are molecular risk factors of COVID-19 related death. The individual risk factor genes or gene sets are potential mediators in disease pathogenesis. These findings allow for better prediction of severe outcome, inform the development of preventative measures to reduce fatality, and inform the evaluation of long-term health impact of COVID-19 in different populations.

## ACKNOWLEDGEMENTS

We thank Dr. Terry Christenson for his professional editing of the manuscript.

### Funding

The authors received no funding for this work.

### Competing Interests

The authors declare that they have no competing interests.

### Author Contributions

- Matthew Lee conceived and designed the experiments, performed the experiments, analyzed the data, prepared figures and/or tables, authored or reviewed drafts of the paper, and approved the final draft.
- Yung Chang conceived and designed the experiments, authored or reviewed drafts of the paper, and approved the final draft.
- Navid Ahmadinejad analyzed the data, prepared figures and/or tables, and approved the final draft.
- Crista Johnson-Agbakwu analyzed the data, authored or reviewed drafts of the paper, and approved the final draft.
- Celeste Bailey analyzed the data, authored or reviewed drafts of the paper, and approved the final draft.
- Li Liu conceived and designed the experiments, performed the experiments, analyzed the data, prepared figures and/or tables, authored or reviewed drafts of the paper, and approved the final draft.

### Data Availability

The data is available at NCBI GEO: GSE63060, GSE110551, GSE12288, GSE42057, GSE30101, GSE104640, GSE135111, GSE37356, GSE110551, GSE93272, GSE65561, GSE37171, GSE21942, GSE28619.

### Supplemental Information

Supplemental information for this article can be found online at http://dx.doi.org/10.7717/peerj.13227#supplemental-information.

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
