# Peer review of "COVID-19 mortality is associated with pre-existing impaired innate immunity in health conditions"

_PeerJ, doi:10.7717/peerj.13227_

## Round 0.1 · original submission · Major Revisions

Please fully respond to the reviewer comments.

Reviewer 1 ·

Basic reporting

1. The present study examines the relationship between COVID-19 mortality and pre-existing impaired innate immunity.
2. Paper reports the analyzed transcriptomics data to find the differentially expressed genes between individuals affected versus unaffected for each pre-existing health condition. Next, the paper examined the relationship between the differentially expressed genes and the increased risk of COVID-19 related death.
3. The paper uses clear, unambiguous, and professional language throughout.
4. In this paper, there is sufficient background information to show how the work fits into the broader field of knowledge.
5. In general, the paper is self-contained and includes all results relevant to the hypothesis.

Experimental design

Strengths:
1. The paper includes clearly describes the introduction, methods, and major findings.
2. The paper performs sound statistical analysis and reports the details enough to replicate the study.
3. Conclusions are supported by the evidence presented in the paper.
4. The paper describes the knowledge gap associated with pre-existing health conditions and COVID-19 mortality. There are many similar publications to this one, but the current paper presents novel results and interpretations of those results.
5. In addition paper includes the "in-silico" validation of the findings using the public dataset.

Validity of the findings

1. Utilizing the enriched gene sets and the R function cnetplot(), the paper describes the network analysis of the gene sets. The results were used to associate the enriched gene sets with the functional labels and clusters. In any case, the results of these analyses should be taken with a grain of salt as they are qualitative. To quantify the strength and statistical validity, quantitative analysis is recommended. Using the NbClust (Charrad et al. 2014) package of R, the optimum number of clusters can be calculated. Using the Goodman-Kruskal-gamma index (Baker 1974), the obtained dendrograms can be compared directly. Therefore, the subjective interpretation of the network analysis can be made objective.
2. Other related literature reports similar findings such as Treskova-Schwarzbac et al., 2021 in BMC Medicine, which, however, limits the impact of the primary findings of this paper.

Additional comments

The paper is well written and follows a logical flow. Many of its methods are sound, and it provides enough details to make it reproducible.

Reviewer 2 ·

Basic reporting

Minor:
Line plots, Fig. 2B and 2D and Fig. 4B: Are the lines connecting each point from each pre-existing health condition? Showing points may be useful to know observed HR and FC.

Experimental design

1. Abstract and Introduction: Please clarify the purpose of the study. I was confused. Was the purpose (1) to identify gene expression patterns that are different between pre-existing health conditions because their HRs varies (Abstract lines 8-9, lines 19-20 and lines 42-44), (2) to identify gene expression patterns that are unique to those who are susceptible to get severe COVID for each pre-existing health conditions, because individuals have a different chance of the outcome (Abstract line 1-2, lines 23-32, lines 48-51, and lines 55-56), or, (3) to identify the pattern of gene expression changes before and after COVID (lines 34-40 and lines 84-94 in Methods)? Due to unclear purpose of the study, following the interpretation of each data analysis in the Results section was difficult.
2. Line 96-: I could not comprehend the interpretation of gene expression fold change (FC) similarity among different pre-existing health conditions. For each pre-existing health condition, FC was calculated using individuals with and without COVID. Then, genes with similar FC among pre-existing health conditions were identified. Biologically, what would similar FC indicate in this case? The comparison of FC among different pre-existing health conditions is confounded. For example, different pre-existing health conditions may have different gene expression patterns without COVID. Then, COVID may result in making similar gene expression levels regardless of the expression level without COVID. So, similar FC among different pre-existing health conditions may have different gene expression levels with COVID, and different FC may have similar gene expression levels with COVID. Clarification of the logic behind this analysis will be helpful.

Validity of the findings

no comment

Additional comments

The authors analyzed hazard ratios of COVID-19 mortality of fourteen pre-existing health conditions. Using transcriptomics data, differentially expressed genes between individuals with and without COVID were identified. They found a correlation between altered gene expression patterns and hazard ratios of COVID-19 mortality. Also, they found that some gene expression pattern differences between with and without COVID were similar among different pre-existing health conditions. They further found that these genes were associated with viral replication and innate immune responses. Their results are very interesting.

---

## Round 0.2 · accepted · Accept

Congratulations. You have addressed all the reviewer comments well.

Reviewer 1 ·

Basic reporting

NA

Experimental design

NA

Validity of the findings

NA

Additional comments

My concerns have been addressed by the authors, and all my suggested changes have been implemented in the paper. There are no further comments.

Reviewer 2 ·

Basic reporting

I appreciate the authors' clarification. I do not have any further concerns.

Experimental design

no comment

Validity of the findings

no comment